# Reducing missed opportunities for vaccination in Mozambique: findings from a cross-sectional assessment conducted in 2017

Bvudzai Priscilla Magadzire  ,[1] Gabriel Joao,[2] Ruth Bechtel,[2] Graça Matsinhe,[3] Laura Nic Lochlainn,[4] Ikechukwu Udo Ogbuanu[4]

¹Global Technical Team, VillageReach, Seattle, Washington, USA
²Mozambique Country Program, VillageReach, Maputo, Mozambique
³Expanded Program on Immunization, Ministry of Health (Ministro da Saude), Maputo, Mozambique
⁴Department of Immunization, Vaccines and Biologicals (IVB), World Health Organization, Geneva, Switzerland

**Correspondence to**
Dr Bvudzai Priscilla Magadzire;
bvudzai.magadzire@villagereach.org

## ABSTRACT

**Objective**  Missed opportunities for vaccination (MOV) are a significant contributor to low vaccination coverage. To better understand the magnitude and underlying causes of MOV among children aged 0–23 months in Mozambique, we conducted an assessment and developed a roadmap for strengthening the country's childhood immunisation programme.

**Setting**  Three provinces in North, South and Central Mozambique.

**Methods**  The assessment applied a mixed-method approach. From a sample of 41 health facilities, we conducted exit interviews with caregivers of children aged 0–23 months (n=546), surveys with health workers (n=223), focus group discussions with caregivers (n=6) and health workers (n=5), and in-depth interviews with health facility managers (n=9). We analysed the data to assess the magnitude of MOV and to identify causes of MOV and ways of preventing them.

**Results**  Vaccination records were available for 538 children. Sixty per cent (n=324) were eligible for vaccination on arriving for their health facility visit. Of these, 76% (n=245) were not fully vaccinated, constituting MOV. Our analysis shows that these MOV were most frequently attributable to practices of caregivers and health workers and also to health systems reasons. Inadequate information about vaccination among both caregivers and health workers, poor or poorly understood health facility practices, inadequate integration of curative and preventative services, inadequate tracking systems to identify children due for vaccination and, less often, limited supply of vaccines, syringes and other related supplies at service points resulted in MOV.

**Conclusions**  The results of the assessment informed the development of roadmaps for reducing MOV that may be applicable to other settings. The global immunisation community should continue to invest in efforts to reduce MOV and thereby make health service visits more effective and efficient for vaccination.

## Strengths and limitations of this study

► Ministry of Health teams were able to use the results immediately to inform the development of roadmaps to reduce missed opportunities for vaccination (MOV).

► The assessment covered a relatively short-time period (10–12 days) and a small sample of health facilities, and only included children with documented vaccinations. Consequently, other reasons for MOV may have been missed.

► As with past MOV assessments, the current assessment was not intended to be nationally representative or representative of any of the three provinces that were sampled. Results may therefore not be generalisable.

► MOV were estimated based only on children with documented vaccination dates extracted from the children's home-based records or from health facility registers. We noted poor vaccination recording practices that may have resulted in many children being undervaccinated or overvaccinated with certain antigens, leading to over-estimation or under-estimation of MOV.

2015.[1] Although several factors may account for this gap, persistently high rates of missed opportunities for vaccination (MOV), particularly in low-income and middle-income countries, are a significant contributor.[2] The WHO defines a MOV as 'any contact with health services by an individual who is eligible for vaccination, which does not result in them receiving all the vaccine doses for which he or she is eligible'.[3–5] It is widely assumed that children who are consistently missed by national immunisation programmes have limited access to health services—one in five children by global estimates, hence a focus on reaching the 'fifth child'. Evidence suggests that a significant proportion of these unvaccinated children may already be accessing treatment and even seeking vaccination services,

## INTRODUCTION

Globally, the proportion of children receiving three diphtheria–tetanus–pertussis routine immunisations is below the global target of 90% and has remained around 85% since

but they are still not receiving all of the recommended routine vaccinations.[3–5]

There is increasing momentum at the global, regional and country levels to address factors that may result in MOV. Accordingly, since 2015 WHO has been working with multiple immunisation partners to improve and scale up its global MOV strategy. This strategy comprises four main components: the *Planning Guide to Reduce Missed Opportunities for Vaccination*,[4] the revised and updated *Methodology for the Assessment of Missed Opportunities for Vaccination*,[5] the *Intervention Guidebook for Implementing and Monitoring Activities to Reduce Missed Opportunities for Vaccination (MOV)*[6] and coordination among immunisation partners and countries.

The MOV strategy aims to answer three essential questions:

► How many opportunities for vaccination are missed at existing vaccination sites?
► Why are opportunities for vaccination missed at these vaccination sites?
► What can be adjusted or done differently (eg, changes in policies, changes in behaviour or structural or organisational changes) to reduce MOV?

By identifying the underlying causes of MOV, countries will be able to tailor health system interventions to reduce MOV, with the goal of improving vaccine coverage, timeliness of vaccination and equity.

The Mozambique Expanded Programme on Immunization (EPI) faces challenges in increasing immunisation coverage. For example, national coverage of the third dose of diphtheria–tetanus–pertussis and the first dose of measles-containing vaccine have remained at 80% and 85%, respectively, since 2015.[7] At the subnational level, many areas of the country have varying degrees of coverage. Approaches to strengthen vaccine delivery—from optimising vaccine supply chains to short-term interventions such as immunisation campaigns—also vary.[8 9] Despite these multistakeholder efforts to address the challenges, new cases of circulating vaccine-derived poliovirus and pockets of unvaccinated children in Mozambique indicate an urgent need to strengthen the EPI.[10]

Consequently, in 2017 Mozambique's Ministry of Health, along with a consortium of partners that included WHO and VillageReach (a non-profit global health organisation), implemented the MOV strategy in Mozambique.[4–6] The objectives were to estimate the prevalence of MOV - the timeliness of vaccination; factors associated with MOV among children aged 0–23 months; and health-worker knowledge, attitudes and practices (KAP), and to formulate recommendations based on these data to improve immunisation policies and practices in Mozambique.

## METHODS
### Study design and setting
The MOV assessment methodology uses a mixed-methods approach that incorporates both quantitative and qualitative tools which allows for triangulation of data. Quantitative data were collected by conducting exit interviews with caregivers and KAP surveys with health workers. In 2017, qualitative data were collected through focus group discussions (FGDs) with caregivers and health workers, as well as through in-depth interviews with health facility managers. The detailed MOV assessment methodology has been described elsewhere.[1 11–15]

The Ministry of Health selected the following three provinces: Niassa (Northern region), Zambezia (Central region) and Maputo (Southern region) for geographical representation and because VillageReach is present in these regions. Two districts were selected in each province. We conducted a desk review using two electronic record systems as data sources: a system for vaccine logistics (Sistema Electrónico de Logística de Vacinas) and a monitoring and evaluation information system (Sistema de Informação de Saúde para Monitoria e Avaliação). We reviewed data for a period of 3 months (April–June 2017), examining the following issues: administrative vaccination coverage, vaccine stock-out rates and functioning of cold-chain equipment (online supplemental table 1). Districts with vaccination coverage greater than 80%, a stock-out rate lower than 10% and more than 95% of fridges functioning were considered 'well performing'. Districts with vaccination coverage less than 50%, a stock-out rate higher than 10% and fewer than 95% of fridges functioning were considered 'poor performing'.

Because health services are decentralised, within each province we selected a combination of comprehensive primary healthcare and tertiary health facilities that serve populations of different socioeconomic status in urban and rural areas.

### Participant selection
Health facilities' staff are made up of a mix of mostly nurses and technicians and a few doctors. For the health-worker KAP surveys and FGDs, we included health workers in the selected health facilities from both preventative and curative departments who were present on the day of the assessment.

Caregivers collaborated with us as key informants. Their views and experiences alongside those of health workers shaped the design of proposed interventions. All caregivers aged at least 18 years exiting the health facility in the selected districts with a child that appeared to be between 0 and 23 months of age were eligible for inclusion in the exit surveys or FGDs. If more than one child accompanied a caregiver, the interviewers were instructed to select one child between 0 and 23 months of age.[5]

### Data collection tools
The generic data collection tools in the WHO MOV methodology guide were translated into Portuguese.[5] All quantitative data were collected on password-protected tablets using survey forms uploaded to the survey software platform (Zegeba AS, Alesund, Norway). The data collectors directly entered data from the vaccination documentation (home-based records or health facility registers)

into the survey platform software, and a photograph of the vaccination documentation was captured for data validation purposes. Qualitative data notes were recorded on paper and subsequently typed up and emailed to the MOV supervision team. Once paper notes were transcribed, and translated into English for analysis, they were destroyed.

## Data collection and analysis

Prior to data collection, we conducted a week-long training with the field team, which comprised of four supervisors and eight data collectors from the Ministry of Health, independent researchers from Eduardo Mondlane University and VillageReach staff. To consolidate classroom training lessons and ensure comfort with electronic data entry of the surveys and photographing vaccination documentation, we piloted the data collection tools in nearby health facilities in Maputo City. Mozambique was the first country to use the Portuguese version of the WHO MOV tools; consequently, after the pilot, the team assessed the accuracy of the translation and amended the tools as necessary.

Field teams were deployed to the three provinces over a 2-week period between October and November 2017. The teams aimed to achieve a daily team quota of 20 exit interviews with caregivers and 10 KAP surveys with health workers at each health facility. Exit interviews with caregivers were conducted in the morning, and health workers were interviewed in the afternoon. Health facility and provincial health officials were purposively selected to participate in in-depth interviews. Although an initial sample of 30 health facilities was selected centrally, team supervisors liaised with the central coordinator to add new sites if the session sizes at the health facility were insufficient to fulfil the daily team quota. Because data were captured primarily with electronic tools (tablets), quantitative data were uploaded daily to a central server and translated into English for analysis.

All quantitative analyses were conducted using Stata (StataCorp. V.2015. Stata Statistical Software: Release 14. College Station, Texas: StataCorp LP). Data analysis focused on estimating the extent of MOV and timeliness of vaccination using information collected from vaccination documentation. As part of the data cleaning process, all vaccination dates entered by the field teams during data collection were validated by reviewing each photo of the vaccination documentation. We further analysed other explanatory and demographic reasons for MOV, the classification of MOV by underlying cause (eg, related to the health system, caregivers or health workers) and estimated the number of eligible doses missed, by antigen. We assessed the timeliness of vaccination doses of all vaccinations indicated on the childrens' vaccination documents using the recorded date of birth, the dates of vaccination and the time intervals for vaccines recommended by the Mozambique EPI (online supplemental table 2).

Using Atlas.ti (ATLAS.ti Scientific Software Development GmbH, V.8), we conducted a thematic analysis of the qualitative data transcripts to explore common, unique and conflicting themes, as well as to draw out illustrative quotes and key conclusions.

We shared the preliminary MOV assessment results with the leadership of the three provinces for verification and consideration to inform the development of future interventions to reduce MOV.

## Ethics approval

The National Bioethics Committee in Mozambique (Comité Nacional de Bioética) granted ethical clearance for this assessment (reference number: 1008). In addition, each province (Niassa, Zambezia and Maputo) granted permission for the MOV assessment to be conducted. Before each interview, respondents were informed that participation was voluntary and optional. They were also informed that their responses would in no way affect their ability to access services or, in the case of health workers, threaten their employment.

# RESULTS

## Respondent characteristics

The field teams conducted the MOV assessment in 41 health facilities, visiting 11 more health facilities than planned due to the small session sizes encountered at some health facilities. The teams carried out exit interviews with 546 caregivers, of whom 99% (n=542) had children aged 0–23 months and 98% (n=538) had vaccination records. For 41% (n=221) of children aged 0–23 months, the purpose of the health facility visit was for a healthy child check-up, followed by 27% (n=145) for medical consultation; 27% (n=143) for vaccination; 3% (n=18) accompanying a caregiver; and 1% for child registration (n=7) or hospitalisation (n=3) (online supplemental table 3).

In terms of age, three-quarters (74%; n=396) of the children were younger than 12 months, and 51% (n=273) were girls (online supplemental table 3). The caregiver literacy rate (measured as reported ability to read and write) was 45% (n=236). Forty-one per cent (n=216) of caregivers had no educational qualifications, while 29% (n=153) had completed primary education and 10% (n=55) had completed secondary school or had higher education table 1.

We also conducted KAP surveys with 223 health workers at the selected health facilities, including staff from preventative and curative service departments. The majority of health workers interviewed were either trained nurses (42%; n=93) or doctors (34%; n=75). The remainder were from other professional cadres, for example, pharmacists or pharmacist technicians (7%; n=16) and laboratory technicians (4%; n=8). Among participating health workers across the three provinces, only 16% (n=49) routinely administer vaccines in EPI clinics. The remainder worked in outpatient departments

**Table 1** Numbers and proportion of missed opportunities for vaccination (MOV) by demographic and other factors in three provinces in Mozambique, 2017 (n=538)

| Demographic or other factor | Overall | |
| --- | --- | --- |
| | No contraindications, and due at least one dose n (%) | MOV n (%) |
| | 324 (60%) | 245 (76%) |
| Sex of child | | |
| Female | 156 (48%) | 118 (76%) |
| Male | 168 (52%) | 117 (70%) |
| Age of child | | |
| <12 months | 263 (81%) | 177 (67%) |
| ≥12 months | 61 (19%) | 58 (95%) |
| Province | | |
| Maputo | 102 (31%) | 76 (74%) |
| Niassa | 114 (35%) | 61 (56%) |
| Zambezia | 105 (33%) | 98 (93%) |
| Reason for visit | | |
| Vaccination | 130 (40%) | 59 (45%) |
| Non-vaccination visit | 194 (60%) | 176 (91%) |
| *Healthy child visit* | *103 (32%)* | *92 (89%)* |
| *Medical consultation* | *74 (23%)* | *69 (93%)* |
| *Accompanying caregiver* | *12 (4%)* | *11 (92%)* |
| *Hospitalisation* | *2 (<1%)* | *2 (100%)* |
| *Child registration* | *2 (<1%)* | *2 (100%)* |
| Do you have a home-based record for this child? | | |
| Yes, and I have it with me today | 319 (98%) | 230 (72%) |
| Yes, but I do not have it with me | 3 (1%) | 3 (100%) |
| No, I do not have a home-based record for this child | 2 (1%) | 2 (100%) |
| Reason not vaccinated today* | | |
| Vaccination was not reason for visit | – | 92 (59%) |
| Child is already fully vaccinated | – | 29 (18%) |
| Health worker said child was not eligible for vaccination | – | 15 (10%) |
| No vaccines in the health facility | – | 12 (8%) |
| Health worker did not tell me about vaccinating child | – | 6 (4%) |
| Other reasons | – | 3 (2%) |

*Respondents allowed to select multiple responses.

or in oral health, general maternal and child health, family planning, nutrition or other specialty services (data not shown). Two-thirds (n=144) of health workers had fewer than 4 years of work experience. Most health workers (83%; n=183) reported they received training in vaccine-preventable diseases during basic preservice professional training (online supplemental table 4). Only 22% (n=48), however, had received additional in-service training. Of those who had received in-service training, 53% (n=24) had been trained in the past year, 20% (n=9) in the past 2 to 3 years and 27% (n=12) 4 or more years ago.

Finally, the field teams conducted six FGD sessions with caregivers, five FGD sessions with health workers and nine in-depth interviews with provincial health officials and health facility managers.

### Frequency and causes of MOV
#### Proportion of children with MOV

The proportion of children with up-to-date vaccinations at the start of their health facility visit was 40% (n=214), including one child who was ineligible for further vaccination because of a reaction to a previous vaccine dose. Among the 324 children eligible for vaccination, 76% (n=245) had an MOV on the day of the assessment (figure 1). Among the 324 children, 52% (n=170) received no vaccine, while 23% (n=75) received at least

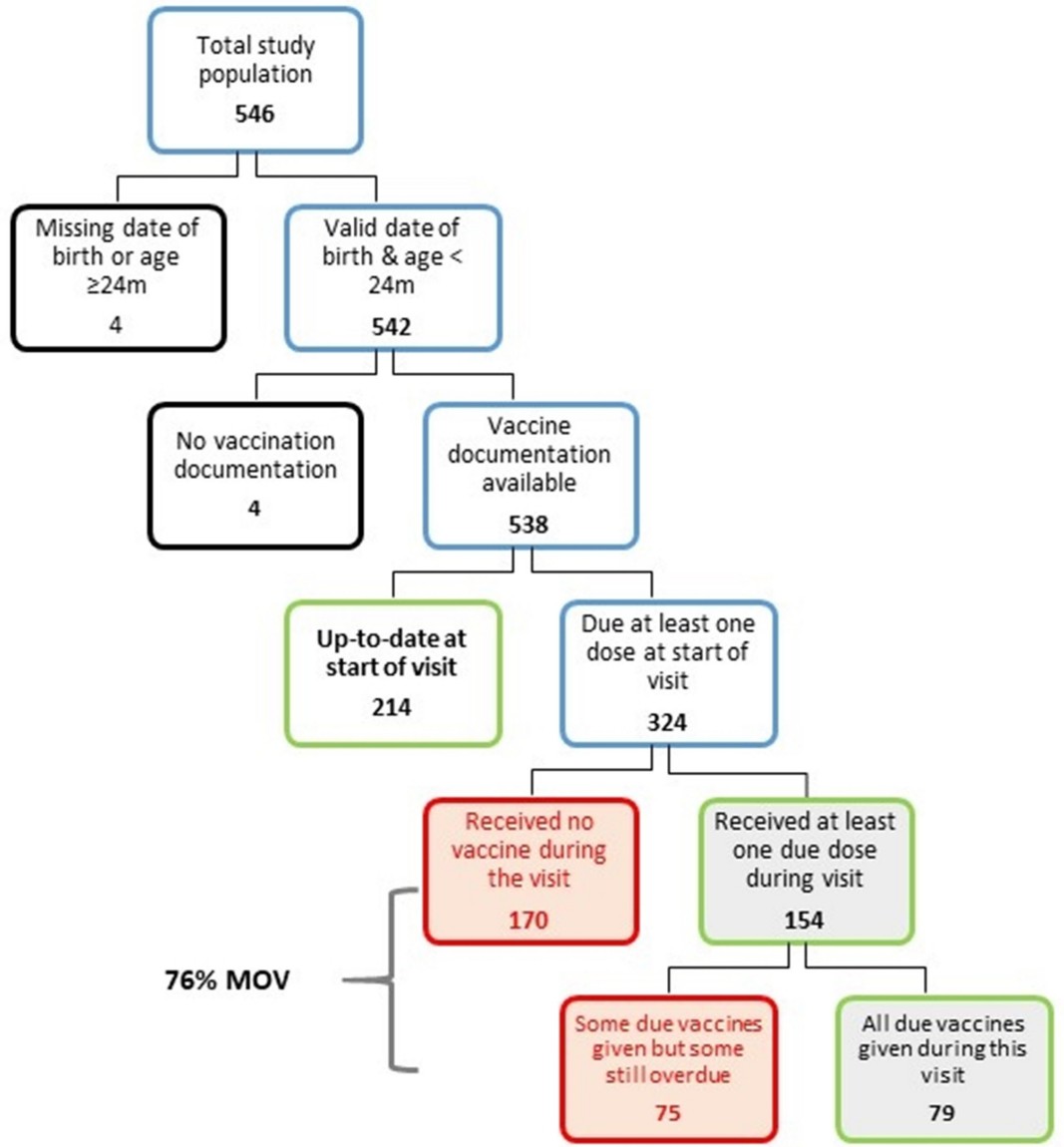

**Figure 1** Flow chart for estimating missed opportunities for vaccination (MOV) in three provinces in Mozambique, 2017.

one dose during the visit, but were still due for other vaccines; 24% (n=79) received all due vaccines.

### Demographic and other factors correlated with MOV

We found the proportion of MOV differed slightly by sex, with 76% (n=118) of girls compared with 70% (n=117) of boys experiencing an MOV. Children aged 12 months or older had a higher proportion of MOV (95%; n=58) compared with children under 12 months (67%; n=177). Children who lived in Zambezia Province had a higher proportion of MOV (93%; n=98) compared with children from Maputo (76%; n=74) and Niassa (56%; n=61).

Forty-five per cent (n=59) of children eligible for vaccination who came to the health facility for vaccination had an MOV, compared with 91% (n=176) of children who were at the health facility for a non-vaccination visit. Among these children, 93% (n=69) had an MOV during their medical consultation, 89% (n=92) during a

healthy child visit and 92% (n=12) when accompanying a caregiver.

When caregivers were asked why their child was not vaccinated during the health facility visit, 59% (n=92) said that the reason for their visit was not for vaccination or they believed their child was already fully vaccinated (18%; n=29). Caregivers also said that health workers told them their child was not eligible for vaccination (10%; n=15) or did not tell them about vaccination during the visit 8% (n=12) or that no vaccines were available in the health facility (8%; n=12) (table 1).

### Eligible vaccine doses missed, by antigen

Of the 851 eligible vaccine doses due, 53% were missed (n=447). BCG (60%; n=40), first dose of oral poliovirus (OPV) (64%; n=39), third dose of OPV (66%; n=38), second dose of measles and rubella (89%; n=33) and

**Table 2** Missed opportunities by vaccine dose and timeliness of vaccine doses administered to surveyed children aged 0–23 months with documented vaccination history, Mozambique 2017

| Vaccine and dose | Eligible doses due n | Eligible due doses | | Timeliness* | | | |
| --- | --- | --- | --- | --- | --- | --- | --- |
| | | Given n (%) | Missed n (%) | Doses children received n | Vaccination given: too early % | Vaccination given: timely % | Vaccination given: delayed % |
| BCG | 67 | 27 (40%) | 40 (60%) | 498 | — | 90% | 10% |
| Zero dose OPV | 32 | 20 (62%) | 12 (38%) | 461 | — | 97% | 3% |
| First dose of penta | 47 | 31 (66%) | 16 (34%) | 459 | 12% | 76% | 12% |
| First dose of OPV | 61 | 22 (36%) | 39 (64%) | 436 | 12% | 71% | 17% |
| First dose of PCV | 48 | 19 (60%) | 29 (40%) | 456 | 11% | 76% | 13% |
| First dose of rotavirus† | 38 | 28 (74%) | 10 (26%) | 419 | 11% | 75% | 14% |
| Second dose of penta | 47 | 39 (83%) | 8 (17%) | 417 | 9% | 75% | 16% |
| Second dose of OPV | 45 | 25 (56%) | 20 (44%) | 385 | 9% | 66% | 25% |
| Second dose of PCV | 48 | 38 (79%) | 10 (21%) | 411 | 9% | 71% | 20% |
| Second dose of rotavirus† | 44 | 40 (91%) | 4 (9%) | 368 | 9% | 50% | 41% |
| Third dose of penta | 49 | 25 (51%) | 24 (49%) | 350 | 6% | 73% | 21% |
| Third dose of OPV | 58 | 20 (34%) | 38 (66%) | 321 | 6% | 64% | 29% |
| Third dose of PCV | 55 | 26 (47%) | 29 (53%) | 340 | 6% | 70% | 24% |
| Inactivated poliovirus | 129 | 12 (9%) | 117 (91%) | 274 | 16% | 65% | 18% |
| First dose of measles | 46 | 28 (61%) | 18 (39%) | 216 | 14% | 67% | 19% |
| Second dose of measles and rubella | 37 | 4 (11%) | 33 (89%) | 23 | 4% | 39% | 57% |
| **Total** | **851** | **404 (47%)** | **447 (53%)** | **5834** | **10%** | **70%** | **21%** |

*Please see online supplemental table 2 for time intervals used in classifying timeliness of vaccination as per the Mozambique immunisation schedule. An em dash (—) indicates no eligible doses were due.
†Rotavirus vaccine given up to 14 weeks of age.
OPV, oral poliovirus vaccine; penta, diphtheria-tetanus-pertussis-hepatitis B–*Haemophilus influenzae* type b vaccine; PCV, pneumococcal conjugate vaccine.

inactivated poliovirus (IPV) vaccine (91% n=117) constituted the highest numbers of missed doses overall (table 2).

## Timeliness of routine immunisation

Based on analysis of all vaccinations indicated on vaccination documents, 70% (n=4102) of the 5834 documented vaccine doses were administered in a timely manner, while 10% (n=558) were too early and 21% (n=1236) were delayed. The vaccine doses most commonly administered too early were IPV (16%; n=45) and the first dose of measles (14%; n=31) (table 2). The most frequent delayed vaccines included the second dose of rotavirus (41%; n=151), the second dose of OPV (25%; n=98), the third dose of OPV (30%; n=94) and the second dose of measles and rubella (57%; n=13). Overall, the pentavalent and OPV vaccine series both showed decreasing timeliness, with doses due at older ages tending to be more delayed (table 2).

We also found 13 children with more than one dose of IPV and 37 children with a third dose of rotavirus recorded in their home-based record (online supplemental table 2).

## Quality, availability and accessibility of vaccines and vaccination services

During the health facility visit, 86% (n=464) of caregivers reported that the health worker asked to see their child's home-based record. Among the 154 caregivers who brought their child for vaccination on the day of the assessment, 96% (n=141) reported being satisfied with the services. However, 64% (n=95) were not told about vaccination reactions their child might have following vaccination, 52% (n=76) were not told what vaccines their child had been given and 23% (n=32) were not informed about the next vaccination date (online supplemental table 3).

Two-thirds (67%; n=33) of health workers who routinely administer vaccines in the EPI clinic felt there were sufficient staff to carry out vaccination activities. However, 22% (n=11) reported having insufficient supplies of specific vaccines, including for BCG, pneumococcal conjugate vaccine, rotavirus and OPV vaccines (online supplemental table 4). Other supplies that were reported to be lacking at the service delivery level included syringes, home-based records and safety boxes.

Physical accessibility (distances and geographical factors) to vaccination services was one of the major barriers identified by both caregivers and health workers in FGDs and in-depth interviews. Some caregivers stated that this is particularly challenging when the child is still very young; they did not want to expose them to harsh weather conditions, or they lacked transportation to the health facility:

> What happens sometimes is that mothers do not appear…when it [health facility] is far away, they begin to have fear. Maybe they just come for those first vaccines until the [baby is] 4 months. (Health worker, FGD, Zambezia Province)

Health workers referred to the important role of mobile brigades (outreach teams) in providing access to vaccination services to the communities. However, resources for mobile brigades, such as transportation and human resources, are limited. As a result, health workers said, maintaining a consistent outreach programme was challenging:

> Another challenge is to cover the entire area of the administrative post including the absentees. There are many absentees. They may give the contact number, but it does not go through. Although some give their addresses, how do we get there? It's hard. Here, many times, we plan mobile brigades, but we can't run them because of insufficient transportation. We only have [an] ambulance and it is doing a lot of work hours and patient transfers. (Health worker, FGD, Maputo Province)

### Knowledge and perceptions of vaccination among caregivers

Although a high percentage (99%; n=533) of caregivers with children aged 0–23 months had their child's home-based record on hand during the health facility visit, the majority of caregivers (75%; n=387) reported not knowing precisely what vaccines their child needed, and 21% (n=112) were unsure about the purpose of vaccination (online supplemental table 3). Of the 78 caregivers who reported knowing the vaccines needed by their child, 36% (n=28) did not know when the vaccines should be given. Gaps in caregiver knowledge on issues related to vaccination were also emphasised in the FGDs:

> We don't know all the vaccines. We know that they protect, yes, but each one protects what? Most of us don't know. Maybe an explanation in terms of what

each of the vaccines do [would help]. And when they increase the vaccines, also explain to us why they are increasing and which vaccines they are adding. (Caregiver, FGD, Niassa Province)

Overall, caregiver responses seemed to reflect that they placed high value on vaccination services; 92% (n=470) stated that children can get sick if they are not vaccinated, and 97% (n=507) reported never having lost a child's home-based record. However, caregivers also raised concerns in FGDs about vaccine acceptability. Caregivers mentioned that the number of vaccinations that children receive is too high:

> The big problem is also the amount of vaccines. Too many vaccines at the same time. There should be a way to reduce the vaccines because children suffer. You see that a child of nine months gets four vaccines in a single day. That's a lot. (Caregiver, FGD, Maputo Province)

### Knowledge and perception of vaccination among health workers

Seventy-eight per cent (n=174) of health workers agreed that their knowledge about vaccines was outdated. A high percentage reported low-grade fever (43%; n=95) and seizures under medical treatment (63%; n=142) as contraindications for any vaccine. Seventy-two per cent (n=152) of health workers also reported that they feared adverse events following immunisation. Only 63% (n=140) of health workers correctly selected all six vaccines that healthy children should receive. Fifty per cent (n=112) of health workers thought that a child's vaccination status should be checked during every type of health service encounter (online supplemental table 4). More than one-third (37%; n=51) suggested that only the health professional responsible for vaccination has an obligation to ask about vaccination status. When health workers were asked why vaccination is incomplete for some children, 56% (n=124) they reported that caregivers' negative perceptions of vaccination are a major contributing factor to children missing vaccinations.

Although 80% (n=177) of health workers did not perceive recording vaccine doses on health facility registers and home-based records as a cause of delayed vaccination (online supplemental table 4), in FGDs they said that they needed improved systems to retrieve children's records quickly and follow-up with children who did not return for vaccine doses:

> Right now what we consider a challenge is actually the children's retention in the vaccination programme, from the first to the last shot. We have books that allow us to follow the child from day one in the same book. Now, the search for defaulters is one of the greatest challenges. (Health worker, FGD, Maputo Province)

Health workers identified three actions as critical for follow-up of defaulting children: home visits, developing a weekly list of children with incomplete vaccinations and contacting caregivers by phone or other means. One of the three provinces was already implementing these interventions, but reported needing more resources to improve intervention effectiveness.

## DISCUSSION

In this first assessment in Mozambique to assess MOV, we have shown that among 60% of children eligible and due for vaccinations at the start of a health facility visit, 76% had an MOV. This proportion of MOV is higher compared with other WHO led MOV assessments from Chad (51%), Malawi (66%), Timor Leste (41%), but similar in proportion to MOV assessments conducted in Kenya (75%) and Burkina Faso (76%).[11–13 15 16] Although 99% of caregivers had vaccination documentation available during the visit, which was the highest percentage observed in an MOV assessment to date, 9 in 10 children (91%) who were at the health facility for a non-vaccination visit had an MOV. We found that the proportion of MOV was higher among girls (76%) compared with boys (70%) and that 95% of children aged 12–24 months who were eligible for vaccination had an MOV. Nine in 10 children (93%) who lived in Zambezia Province had an MOV compared with 74% of children in Maputo and 56% of children in Niassa.

Evidence from a growing body of WHO-supported MOV assessments suggests a significant proportion of children with access to health services experience MOV.[3 11] We found that even though children are accessing health services, the health system continues to miss opportunities to vaccinate eligible children for a variety of reasons. In addition, more than nine out of ten caregivers whose child had an MOV said that the reason for the health facility visit was not for vaccination. This finding provides evidence that there is insufficient screening and referral of children eligible for vaccination. As such, it highlights a need for improved integration between preventative and curative services. The other most significant factors for missing eligible children included a lack of adequate systems and practices for tracking children's vaccination status at the health facility level, inadequate caregiver knowledge and, less often, vaccine stock-outs or being refused services for coming on the wrong day.

Having a vaccination schedule that includes vaccines in the second year of life can often lead to a reduction in MOV, as it provides another opportunity for a child to receive catch-up vaccinations. In 2015, Mozambique introduced a second dose of measles and rubella at 18 months.[17] However, we found that almost all children aged 12–24 months who were eligible for vaccination had an MOV. In order to benefit from this contact point, the delivery platform for vaccines in the second year of life must be further strengthened in Mozambique.

During brainstorming and debriefing sessions at the provincial and national levels with EPI staff and immunisation stakeholders, the key results were linked with actionable steps to reduce MOV and improve the EPI overall. Stakeholders deliberated on interventions that were, in their view, feasible in the short term. Provinces drafted action plans for reducing MOV as part of the 2019 EPI plans. Some of the themes that emerged during the brainstorming sessions follow. However, it must be noted that implementation of these plans was affected by Cyclone Idai and Cyclone Kenneth in early 2019, which had devastating consequences for Mozambique.[18]

### Having available supplies in health facilities is necessary but not sufficient

Lack of vaccines was the least reported barrier to vaccination. The government has worked with organisations such as VillageReach over many years to improve vaccine supply in both Maputo and Niassa,[8 9 19] so to some extent this finding reflects the positive impact of those efforts. The few cases of stock-outs were likely linked to national or even global stock-outs of BCG, rotavirus and IPV vaccines throughout that year, which affected their availability at EPI clinics.[20] It was striking, however, that even when vaccines were available, patients were sometimes turned away because they came on the wrong day, or a health worker could not confirm the child's eligibility.

### Distance and transport limit access

Access to vaccination due to long distances and poor transport infrastructure was a major barrier identified by both caregivers and health workers during FGDs. Transport is doubly challenging. It limits the ability of health workers to provide outreach services, which in many countries cannot be implemented as planned. It also limits access for many caregivers who must bring children over long distances to reach health services. Extending the reach of health facilities by implementing plans for more frequent outreach or mobile services, particularly in remote areas or in areas with vulnerable populations, must therefore be prioritised.

### Knowledge of vaccination among caregivers, health workers and community leaders must be improved

Our evidence shows that both caregivers and health workers have suboptimal understanding of recommended immunisation policies and practices. During validation of dates based on photos of home-based records, we noted poor vaccination recording practices. Consequently, both health workers and caregivers did not know whether a child has received all due vaccines. To reduce MOV, caregivers need to understand the diseases that each vaccine protects against, the vaccination schedule, the value of vaccination, the importance of retaining the home-based record and potential adverse events following immunisation and how to manage them. Given the low level of literacy among many caregivers, health facilities need materials and training appropriate for improving awareness through relevant information despite limited literacy. Health workers

need to understand their own responsibility for both promoting vaccination and identifying available opportunities, based on practical knowledge of the vaccination schedule. Engaging local radio and community leaders in disseminating information about the importance of vaccination and about relevant aspects of immunisation will help create a more informed environment for both caregivers and health workers.

### Health facilities need better strategies for managing data, electronic registration and retention

Health workers in both preventive and curative service departments should be able to quickly and easily check the vaccination eligibility and status of each child who presents at the health facility. Most health facilities currently use paper-based immunisation registers, from which it is difficult to access information. Well-designed digital registers are needed, and health workers should be better trained to use them effectively. Techniques for designing and managing electronic databases can now minimise errors in entering data, detect and enable correction of discrepancies, and facilitate quick and easy queries, so that health facilities can identify children who are due for vaccination. Aims of effective health data monitoring include tracking children over the course of an immunisation schedule and identifying dropouts. However, this technology would require significant investment for equipment, training, electricity infrastructure and long-term maintenance. In the meantime, efforts to maximise the design and use of paper-based records must continue to be emphasised. Furthermore, ensuring that caregivers always bring home-based records to each health facility visit and regularly checking children's home-based records would facilitate cross-checking clinical and home-based records. Implementing such practices should in turn greatly reduce MOV and the morbidity and mortality of vaccine-preventable diseases.

### Limitations

As with past MOV assessments,[11–13 15 16] this study was not intended to be nationally representative or representative of any of the three provinces that were assessed. It should be regarded, however, as a programme assessment to identify areas which can be improved to reduce MOV. The assessment was carried out over a relatively short time period (10–12 days), in a small sample of health facilities. The estimation of MOV was limited to children with documented vaccination dates extracted from children's home-based records or from health facility registers. As a result, other reasons for MOV may have been missed. In addition, by validating vaccination documentation, we noted poor vaccination recording practices in home-based records. These practices could have resulted in many children being undervaccinated or overvaccinated with certain antigens, leading to over-estimation or underestimation of MOV.

### CONCLUSION

What we learnt in Mozambique confirms a growing global understanding that MOV continue to occur within health facilities. WHO emphasises that in order to reduce MOV, diligent efforts are required in day-to-day practices of health facilities to screen and refer children for vaccination. The results discussed in this report provide evidence and guidance for creating a roadmap for reducing MOV in Mozambique. Our discussions with stakeholders at the provincial and national levels indicate a willingness and commitment to reduce MOV. The country now needs to realise its potential to ensure supplies are available to all populations, improve the timeliness of vaccination services, improve the efficiency of health service delivery and promote synergies between preventive and curative services at the health facility level.

The leadership by provinces in developing tailored action plans and interactions at the national level are impressive. But funding and staff support for enhancing technical expertise and partners are critical. Based on our findings, immunisation stakeholders in Mozambique should proceed with planning and action with the goal of maximising every opportunity to ensure children are fully immunised.

**Acknowledgements** The authors wish to thank the Ministry of Health (Ministerium da Saudi de Moçambique) management and province staff, researchers from Eduardo Mondlane University and partners for supporting the MOV assessment. We also wish to thank the caregivers and health workers who took the time to participate in the MOV assessment. Finally, we would also like to acknowledge the input of Melissa West (VillageReach) during the early phases of manuscript development and Stephanie Shendale (WHO) during the planning phases of the MOV assessment.

**Contributors** All authors conceptualised the study, LNL and IUO developed the methodology, BPM and GJ led data collection and conducted qualitative data analysis and LNL conducted quantitative data analysis. All authors reviewed and validated the results. BPM developed the first-draft manuscript all authors reviewed and approved the manuscript. BPM acts as guarantor for this article.

**Funding** The Bill and Melinda Gates Foundation (OPP1146114) and the WHO supported this study.

**Competing interests** None declared.

**Patient consent for publication** Not applicable.

**Ethics approval** This study involves human participants and was approved by The National Bioethics Committee in Mozambique (Comite Nacional de Bioeetica) granted ethical clearance for this assessment (Reference Number:1008). In addition, each province (Niassa, Zambezia and Maputo) granted permission for the MOV assessment to be conducted. Before each interview, respondents were informed that participation was voluntary and optional. They were also informed that their responses would in no way affect their ability to access services or in the case of health workers threaten their employment. Participants gave informed consent to participate in the study before taking part.

**Provenance and peer review** Not commissioned; externally peer reviewed.

**Data availability statement** Data are available upon reasonable request. All data relevant to the study are included in the article or uploaded as supplementary information. Additional data are available upon reasonable request.

terminology, drug names and drug dosages), and is not responsible for any error and/or omissions arising from translation and adaptation or otherwise.

**ORCID iD**
Bvudzai Priscilla Magadzire http://orcid.org/0000-0002-4164-6233

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
