## [Reviewer comments · BMJ Open]

ARTICLE DETAILS

TITLE (PROVISIONAL)	Reducing missed opportunities for vaccination in Mozambique: Findings from a cross-sectional assessment conducted in 2017
AUTHORS	Magadzire, Bvudzai; Joao, Gabriel; Bechtel, Ruth; Matsinhe, Graça; Nic Lochlainn, Laura; Ogbuanu, Ikechukwu Udo

VERSION 1 – REVIEW

REVIEWER	Amir Mohareb Massachusetts General Hospital, Division of Infectious Diseases
REVIEW RETURNED	15-Mar-2021

GENERAL COMMENTS	I am grateful for the opportunity to read this revised manuscript. I want to emphasize once more that this manuscript includes lots of valuable findings that can potentially be useful for many clinicians, researchers, and public health personnel in other low- and middle-income countries. The authors should be commended for their hard work in this field. However, the writing is still not very clear, and I think this limits the utility and impact of this manuscript. I appreciate that the authors considered and responded to each item for review. They did address many of my previous concerns, particularly with the previously inconsistent (or unclear) quantitative results. However, in other cases, they responded to the review in their Response to Review letter, but did not make an edit or revision to the actual manuscript. I will limit my review here to just a few of those points, and I will defer to the editor on the remainder of manuscript and its appropriateness for the journal audience. For example, I still feel like the section on Setting is lacking detail. This section is important because it will determine how we should interpret the results and apply them to other low- and middle-income settings. The authors' revision here simply states that these facilities serve a variety of populations. I would encourage including details and specificity about the size, capacity, and catchment area of the facilities included in this sample, as I mentioned in my original review. With regards to describing the study outcome in Methods (review #7), I think some of the information the authors included in this Response to Review letter would be useful to put in the manuscript. The current "data analysis" section of the Methods does not actually include this level of detail, or it is not clear. I would still like to see this data used to address some of the questions I raise in review item #12. I still feel like the writing relies on abbreviations that are specific to this field and methodology and may not be familiar to the typical reader of BMJ Open (e.g., KAP,
---

	FGD, EPI, HBR, etc.). I still would like to see the findings of this study compared or applied to other low- and middle-income countries. I recognize that all vaccination programs are different; the authors are experts in their field writing to a general medical audience (this journal is not a sub-specialty journal) so the writing could better target the general reader.
--	--

VERSION 1 – AUTHOR RESPONSE

The writing is still not very clear, and I think this limits the utility and impact of this manuscript. We got a technical copy editor to revise the entire manuscript for clarity and consistency.

The section on Setting is lacking detail. This section is important because it will determine how we should interpret the results and apply them to other low- and middle-income settings. The authors' revision here simply states that these facilities serve a variety of populations. I would encourage including details and specificity about the size, capacity, and catchment area of the facilities included in this sample, as I mentioned in my original review.

We have added a new paragraph under “Study design and Setting” to provide more detail on the setting, populations served and their proximity to health facilities and some information on the staff cadre.

With regards to describing the study outcome in Methods (review #7), I think some of the information the authors included in this Response to Review letter would be useful to put in the manuscript. The current "data analysis" section of the Methods does not actually include this level of detail, or it is not clear.

The information in the response number 7 is similar to that in the text. However, we have reviewed the data analysis section once more to make this section clear.

I would still like to see this data used to address some of the questions I raise in review item #12.

As mentioned in the previous response to the reviewers’ comment 12, the MOV methodology were developed to prioritize “identifying” and “advocating” MOV as a problem rather than calculating a statistically precise “estimate”. Also, when examining those with MOV, the numbers are not sufficient to carry out statistical testing. For the remainder of the questions raised by the reviewers, these areas were highlighted as reasons for MOV, even without having to conduct further statistical analyses. We proposed recommendations to address these issues identified, which we hope will lead to improved immunization service delivery.

I still feel like the writing relies on abbreviations that are specific to this field and methodology and may not be familiar to the typical reader of BMJ Open (e.g., KAP, FGD, EPI, HBR, etc.).

Regarding the abbreviations, we had a technical copy editor revise the entire manuscript. They edited for clarity and consistency, checked the in-text reference citations against the actual list to be sure they matched, spot-checked the numbers mentioned in the text against those in the tables, checked the figure and table callouts. Regarding the point about abbreviations, if an abbreviation was used only once or twice, the editor used only the spelled out form. They allowed KAP and FGD after defining them at first mention because they are used so often that they are easy to keep in mind. However, they spelled out “home-based records” because that one is less familiar.

I still would like to see the findings of this study compared or applied to other low- and middle-income countries.

We have added the following sentence in our discussion comparing the MOV findings from Mozambique to other recent MOV assessments from other low- and middle-income countries. “In this first assessment in Mozambique to assess MOV, we have shown that among 60% of children eligible and due for vaccinations at the start of the a health facility visit, 76% had a MOV. This proportion of MOV is higher compared to other recent WHO led MOV assessments from Chad (51%), Malawi (66%), Timor Leste (41%), but similar in proportion to MOV assessments conducted in Kenya (75%) and Burkina Faso (76%)”.

I recognize that all vaccination programs are different; the authors are experts in their field writing to a general medical audience (this journal is not a sub-specialty journal) so the writing could better target the general reader.

As mentioned in the response regarding copy editing, the technical editor revised the manuscript to ensure that it was clear and readable to the general audience. Although we recognize that this journal is not a sub-specialty journal, MOV is a global strategy and BMJ recently published an article by colleagues who examined “Determining the burden of missed opportunities for vaccination among children admitted in healthcare facilities in India: a cross-sectional study”
<https://bmjopen.bmj.com/content/11/3/e046464.long>